# Classic Chromophobe Renal Cell Carcinoma Incur a Larger Number of Chromosomal Losses Than Seen in the Eosinophilic Subtype

**DOI:** 10.3390/cancers11101492

**Published:** 2019-10-03

**Authors:** Riuko Ohashi, Peter Schraml, Silvia Angori, Aashil A. Batavia, Niels J. Rupp, Chisato Ohe, Yoshiro Otsuki, Takashi Kawasaki, Hiroshi Kobayashi, Kazuhiro Kobayashi, Tatsuhiko Miyazaki, Hiroyuki Shibuya, Hiroyuki Usuda, Hajime Umezu, Fumiyoshi Fujishima, Bungo Furusato, Mitsumasa Osakabe, Tamotsu Sugai, Naoto Kuroda, Toyonori Tsuzuki, Yoji Nagashima, Yoichi Ajioka, Holger Moch

**Affiliations:** 1Histopathology Core Facility, Faculty of Medicine, Niigata University, Niigata 951-8510, Japan; riuko@med.niigata-u.ac.jp (R.O.); ajioka@med.niigata-u.ac.jp (Y.A.); 2Department of Pathology and Molecular Pathology, University and University Hospital Zurich, CH-8091 Zurich, Switzerland; Peter.Schraml@usz.ch (P.S.); Silvia.Angori@usz.ch (S.A.); Aashil.Batavia@usz.ch (A.A.B.);; 3Division of Molecular and Diagnostic Pathology, Graduate School of Medical and Dental Sciences, Niigata University, Niigata 951-8510, Japan; 4Department of Biosystems Science and Engineering, ETH Zurich, 4058 Basel, Switzerland; 5Department of Pathology and Laboratory Medicine, Kansai Medical University, Hirakata, Osaka 573-1010, Japan; ohec@hirakata.kmu.ac.jp; 6Department of Pathology, Seirei Hamamatsu General Hospital, Hamamatsu, Shizuoka 430-8558, Japan; otsuki@sis.seirei.or.jp; 7Department of Pathology, Niigata Cancer Center Hospital, Niigata 951-8566, Japan; takawa@niigata-cc.jp; 8Department of Pathology, Tachikawa General Hospital, Nagaoka, Niigata 940-8621, Japan; h-kobayashi15@tatikawa.or.jp; 9Department of Pathology, Gifu University Hospital, Gifu 501-1194, Japan; hern@live.jp (K.K.); tats_m@gifu-u.ac.jp (T.M.); 10Department of Pathology, Niigata City General Hospital, Niigata 950-1197, Japan; shibuya2u@hosp.niigata.niigata.jp; 11Department of Diagnostic Pathology, Nagaoka Red Cross Hospital, Nagaoka, Niigata 940-2085, Japan; usuda@nagaoka.jrc.or.jp; 12Division of Pathology, Niigata University Medical & Dental Hospital, Niigata 951-8520, Japan; umezu@med.niigata-u.ac.jp; 13Department of Anatomic Pathology, Graduate School of Medicine, Tohoku University, Sendai, Miyagi 980-8575, Japan; ffujishima@patholo2.med.tohoku.ac.jp; 14Cancer Genomics Unit, Clinical Genomics Center, Nagasaki University Hospital, Nagasaki 852-8501, Japan; befurusato@me.com; 15Department of Pathology, Graduate School of Biomedical Sciences, Nagasaki University, Nagasaki 852-8501, Japan; 16Department of Molecular Diagnostic Pathology, School of Medicine, Iwate Medical University, Yahaba-cho, Shiwa-gun, Iwate 028-3695, Japan; mosakabe@iwate-med.ac.jp (M.O.); tsugai@iwate-med.ac.jp (T.S.); 17Department of Diagnostic Pathology, Kochi Red Cross Hospital, Kochi 780-8562, Japan; kurochankochi@yahoo.co.jp; 18Department of Surgical Pathology, Aichi Medical University Hospital, Nagakute, Aichi 480-1195, Japan; tsuzuki@aichi-med-u.ac.jp; 19Department of Surgical Pathology, Tokyo Women’s Medical University Hospital, Shinjuku-ku, Tokyo 162-8666, Japan; nagashima.yoji@twmu.ac.jp

**Keywords:** chromophobe renal cell carcinoma, pale cell, eosinophilic variant, chromosomal loss, copy number analysis, renal cell carcinoma

## Abstract

Chromophobe renal cell carcinoma (chRCC) is a renal tumor subtype with a good prognosis, characterized by multiple chromosomal copy number variations (CNV). The World Health Organization (WHO) chRCC classification guidelines define a classic and an eosinophilic variant. Large cells with reticular cytoplasm and prominent cell membranes (pale cells) are characteristic for classic chRCC. Classic and eosinophilic variants were defined in 42 Swiss chRCCs, 119 Japanese chRCCs and in whole-slide digital images of 66 chRCCs from the Cancer Genome Atlas (TCGA) kidney chromophobe (KICH) dataset. 32 of 42 (76.2%) Swiss chRCCs, 90 of 119 (75.6%) Japanese chRCCs and 53 of 66 (80.3%) TCGA-KICH were classic chRCCs. There was no survival difference between eosinophilic and classic chRCC in all three cohorts. To identify a genotype/phenotype correlation, we performed a genome-wide CNV analysis using Affymetrix OncoScan^®^ CNV Assay (Affymetrix/Thermo Fisher Scientific, Waltham, MA, USA) in 33 Swiss chRCCs. TCGA-KICH subtypes were compared with TCGA CNV data. In the combined Swiss and TCGA-KICH cohorts, losses of chromosome 1, 2, 6, 10, 13, and 17 were significantly more frequent in classic chRCC (*p* < 0.05, each), suggesting that classic chRCC are characterized by higher chromosomal instability. This molecular difference justifies the definition of two chRCC variants. Absence of pale cells could be used as main histological criterion to define the eosinophilic variant of chRCC.

## 1. Introduction

Chromophobe renal cell carcinoma (chRCC) is a distinct histological entity of renal cell carcinoma (RCC) described by Thoenes et al. [1] in 1985. chRCC accounts for approximately 5–7% of RCC [2,3,4]. Thoenes et al. used the term chromophobe cell for larger cells with reticular, but not clear cytoplasm and prominent cell membranes (plant cell-like) [1,2]. Three years later, these authors described eosinophilic cells with smaller size and with fine oxiphilic granularity as a second cell component of chRCC [3]. Crotty et al. used the term pale cell instead of the formerly used term chromophobe cell and considered pale cell and eosinophilic cell [5] as two main cell types in chRCC. Several ultrastructural studies showed that pale cells are characterized by numerous cytoplasmic microvesicles, a feature probably related to defective mitochondrial development, whereas mitochondria are abundant in eosinophilic cells [2,6,7,8].

Most chRCCs consist of both cell types, which are typically mixed, with eosinophilic cells usually arranged at the center and pale cells usually arranged at the periphery of the sheets or nests [2]. The 2016 World Health Organization (WHO) renal tumor classification acknowledges an eosinophilic variant of chRCC “that is sometimes difficult to distinguish from renal oncocytoma” [3,4,5,9,10] but there are no exact diagnostic criteria to classify an eosinophilic chRCC. 

Previous studies have demonstrated losses of one copy in many chromosomes, especially in chromosomes 1, 2, 6, 10, 13, 17, 21 and sex chromosome in the majority of chRCCs (~71%). Losses of chromosome 1 and sex chromosome have been also reported in oncocytoma [4,9,11,12,13,14,15]. Recently, mutation of *TP53*, *PTEN*, *HNF1B* were observed in chRCCs [13,16,17]. Previous studies also disclosed frequent somatic mitochondrial DNA mutations in oncocytoma [13,18,19] and chRCCs [9,20], but a clear genotype/phenotype correlation has never been described in chRCC.

In this study, we analyzed the histopathological variants of chRCCs in 42 Swiss, 119 Japanese and in whole-slide digital images of 66 chRCCs from The Cancer Genome Atlas (TCGA) Kidney Chromophobe (KICH) dataset. Further, we utilized single-nucleotide polymorphism (SNP) arrays to assess genome-wide copy number variation (CNV) and correlated CNV to the histological variants in the Swiss and the TCGA-KICH data [13].

## 2. Results

### 2.1. Swiss Cohort

There were 22 of 33 (66.7%) classic chRCCs with typical voluminous pale cells in the Swiss cohort (Table 1). Tumors with and without pale cell are shown in Figure 1. There was no association between the chRCC subtypes with age, sex, and pT stage (Table 2). Molecular analysis using the OncoScan^®^ CNV Assay (Affymetrix/Thermo Fisher Scientific, Waltham, MA, USA) revealed loss of part (>5% of gene loci) or the entirety of chromosome 1, 2, 6, 10, 13, 17, 21, and sex chromosome in the majority of cases (Figure 2 and Table 2). Among these chromosomal losses, chromosome 1 was most affected (32/33, 97.0%). Chromosome 2 (24/33, 72.7%), 6 (26/33, 78.8%), 10 (21/33, 63.6%), 13 (23/33, 69.7%), 17 (25/33, 75.8%), and 21 (17/33, 51.5%) were less frequently altered (Table 2). 

The Correlation between chromosomal losses and clinicopathological features of 33 chRCCs are summarized in Table 2. Classic chRCC showed significantly more chromosome 2 (*p* < 0.05), and chromosome 6 losses (*p* < 0.01) than eosinophilic RCC (Table 2). Among 22 classic chRCCs with pale cells, 19/22 (86.4%) showed chromosome 2 loss, 21/22 (95.5%) chromosome 6 loss, 16/22 (72.7%) chromosome 10 loss, 17/22 (77.3%) chromosome 13 loss, and 19/22 (86.4%) chromosome 17 loss.

### 2.2. TCGA Cohort

53 of 64 (80.3%) were classic chRCCs (Table 1 and Table 2). Classic chRCCs from the Cancer Digital Slide Archive are shown in Figure 3. The publicly available copy number variation (CNV) analysis data of TCGA-KICH dataset revealed losses of chromosomes 1, 2, 6, 10, 13, 17, and 21 in the majority of chRCC as previously reported [13]. The CNV data are summarized in Appendix A. 

Classic chRCC showed significantly more chromosome 1, 2, 6, 10, 17 copy number (CN) losses (*p* < 0.01) and chromosome 13 CN loss (*p* < 0.05) (Appendix A). When the Swiss and TCGA-KICH cohorts were combined, losses of chromosome 1, 2, 6, 10, 13 and 17 were significantly more frequent in classic chRCC with pale cells (*p* < 0.05, each) (Table 2).

### 2.3. Chromophobe Renal Cancer Subtype and Survival 

Higher pT stage (pT3–4 vs. pT1–2) and higher pN stage (pN1–2 vs. pN0) were significantly associated with worse survival by log-rank test (Figure 4a,b) and univariate Cox regression analysis (Table 3) in the Swiss-TCGA-Japanese cohort. There was no overall survival (OS) difference between classic and eosinophilic chRCC subtypes in the three independent cohorts of the Swiss (42 cases), TCGA-KICH (64 cases) and the Japanese (119 cases) nor in the Swiss-TCGA-Japanese combined cohort (Figure 4c). Multivariate Cox regression analysis, including pT stage (pT3–4 vs. pT1–2), pN stage (pN1–2 vs. pN0), WHO/ISUP grade (Grade 3/4 vs. Grade 2), and chRCC subtype showed that pT stage and pN stage were independent prognostic factors for OS, whereas no prognostic impact of the chRCC subtype or WHO/International Society of Urological Pathology (ISUP) grade was observed (Table 3).

### 2.4. Chromosomal Copy Number Variation and Survival

Both, CN data and survival data were available from 30 Swiss chRCCs and 64 chRCCs from TCGA-KICH cohort. In the Swiss cohort, neither CN losses of each chromosome 1, 2, 6, 10, 13, 17, 21 in single analysis nor CN loss of any chromosome among chromosome 1, 2, 6, 10, 13, 17, 21 were associated with worse survival by log-rank test and univariate Cox regression analysis (Table 4). In the combined Swiss-TCGA cohort, only chromosome 21 CN loss was associated with shorter overall survival, whereas all other chromosomes were not associated with survival (Figure 5a and Table 4). Multivariate analysis showed that pT stage was the only independent prognostic factor for OS whereas no association was found between OS and CN loss of chromosome 21 or CN loss of any other chromosome among chromosome 1, 2, 6, 10, 13, 17, 21 (Table 5). Importantly, chRCCs without any CN loss of chromosome 1, 2, 6, 10, 13, 17, 21 groups revealed 100% survival in the combined Swiss/TCGA-KICH cohorts (Figure 5b and Table 4). 

## 3. Discussion

In our study, we used the absence of voluminous pale cells to define eosinophilic chRCC. Using this definition, classic chRCC is associated with significantly more frequent losses of chromosomes 1, 2, 6, 10, 13, and 17. 

Various cytogenetic, comparative genomic hybridization, and recent molecular studies have confirmed the very unique and characteristic genotype with multiple chromosomal losses in chRCC [4,9,11,12,13,14,15]. However, previous attempts to correlate histological variants of chRCCs with a specific genotype have failed. More than 10 years ago, Brunelli et al. analyzed classic and eosinophilic chRCCs by fluorescence in situ hybridization, but they have not observed different frequencies of chromosomal 2, 6, 10, and 17 losses [11]. This is in contrast to our OncoScan results with more chromosomal CNV in classic than in eosinophilic chRCC. Our results are in line with a TCGA-KICH study by Davis et al., demonstrating in almost all classic chRCC there are characteristic chromosomal copy-number losses, whilst approximately 50% of all eosinophilic chRCC (9 of 19) experienced no chromosomal copy-number alterations [13]. Recently, Trpkov et al. proposed low-grade oncocytic tumors (LOT) as an emerging renal tumor entity [22]. They argue that LOT lacks multiple chromosomal losses and gains, and exhibits indolent clinical behavior. This tumor does not fit completely into either oncocytoma or eosinophilic chRCC, despite showing some similarities with both entities. Further studies are warranted to proof that LOT potentially represents a distinct type of tumor or if they should be regarded as variant of eosinophilic chRCC.

During our study design and the re-evaluation of histological slides for this study, we realized that there are no stringent diagnostic criteria to classify eosinophilic chRCC. The current 2016 WHO classification states that eosinophilic chRCC is almost purely composed of eosinophilic cells and that the majority of cells should be eosinophilic cells [2]. Given this lack of exact criteria, we decided to use the complete absence of pale cells as definition for eosinophilic chRCC, because pale cells are easily identifiable and can be clearly separated from eosinophilic cells. 

As a consequence of this lack of stringent criteria for subtyping chRCC, distribution of chRCC variants varies extremely between different studies [2,3,10,11,23,24]. Davis et al. recently classified the TCGA-KICH tumors as classic and eosinophilic variants [13]. Our evaluation of TCGA-KICH digital whole slide images for chRCC only partially matched his classification of classical and eosinophilic variants, which can be explained by our more conservative cut-off to define eosinophilic chRCC (complete absence of pale cells). 

Interestingly, there were no survival differences between eosinophilic and classic chRCC in 3 cohorts from TCGA, Japan and Europe. Given the morphological overlap between eosinophilic chRCC and benign oncocytoma, one could assume that eosinophilic chRCC have a better prognosis than classic chRCC. The prognostic similarity between eosinophilic and classic chRCC further underlines the importance to clearly separate eosinophilic chRCC from oncocytoma. Most importantly, classic chRCC had significantly more losses of chromosome 2 and 6 in the Swiss tumors and more losses of chromosome 1, 2, 6, 10, 13, and 17 in the TCGA dataset. Swiss classic chRCC showed only a trend to more chromosome 1, 10, 13, or 17 losses, probably due to the lower number of cases (Table 2). Almost all eosinophilic and all classic chRCC (91–100%) revealed chromosome 1 loss, suggesting that chromosome 1 loss may be an early event in chRCC tumorigenesis. Chromosome 1 losses have even been identified in oncocytoma [9,11,12,15]. This could be due to the misclassification of eosinophilic chRCCs as oncocytoma, but it is also tempting to speculate that there is a stepwise progression from oncocytoma to eosinophilic or classic chRCC with chromosome 1 loss as a genetic driver. 

Treatment outcomes are poorly characterized in patients with metastatic chRCC. This is a consequence of rare metastasis of this subtype. Patients with metastatic chRCC can be treated with tyrosine kinase inhibitors. It has been recently shown that outcomes between metastatic chRCC and clear cell renal cell carcinoma (ccRCC) are similar when treated with conventional targeted therapies [25]. In addition, chRCC has to activate mutations in phosphatase and tensin homolog (PTEN)-phosphatidylinositol 3-kinase (PI3K)/Protein Kinase B (Akt)/mammalian target of the rapamycin (mTOR) pathway [13], which would result in an appropriate target for an mTOR inhibitor. Genomic instability, including whole-chromosome aneuploidy, is a hallmark of human cancer, but the level of chromosomal losses in chRCC is unique. We have recently identified *SF3B1* on chromosome 2 as a Copy-number alterations Yielding Cancer Liabilities Owing to Partial losS (CYCLOPS) gene with a highly significant positive correlation to hypoxia-inducible factor-1α (HIF1α) [26]. It is therefore tempting to speculate that an Splicing factor 3B subunit 1 (SF3B1)/HIF1α pathway with potential therapeutical relevance exists in chRCC. 

Due to the unique genomic background, chRCC should be enrolled in separate clinical trials to measure outcomes. However, chRCC is mostly included in clinical trials together with other non-clear cell RCCs. Accurate classification of metastatic lesions is therefore important as chRCC should be managed with different treatment algorithms. Kouba et al. have recently demonstrated that cytogenetics, showing multiple genetic losses is an additional tool in a metastatic RCC lesion for differential diagnosis of the primary [27]. Our own data show that chRCC without chromosomal losses have an indolent behavior. Therefore, analysis of chromosomal losses by fluorescence in situ hybridization (FISH) or other technologies could be used to assess the behavior of chRCC in organ-confined tumors or to better characterize metastatic lesions of RCC.

## 4. Materials and Methods 

### 4.1. Swiss Patients

chRCC patients were identified from the files of the Department of Pathology and Molecular Pathology of the University Hospital Zurich between 1993 and 2013. Our retrospective study fulfilled the legal conditions according to Article 34 of the Swiss Law “Humanforschungsgesetz (HFG)”, which, in exceptional cases, allows the use of biomaterial and patient data for research purposes without informed consent, if i) it is impossible or disproportionately difficult to obtain patient consent; ii) there is no documented refusal; iii) research interests prevail the individual interest of a patient. Law abidance of this study was reviewed and approved by the ethics commission of the Canton Zurich (KEK-ZH-Nr. 2014-0604 on 1st April 2015; PB_2016-00811 on 22nd February 2016). This study was conducted in accordance with the Declaration of Helsinki. The demographic and clinicopathological characteristics for 42 chRCCs with clinical data are summarized in Table 1.

All tumors were reviewed by two pathologists (R.O. and H.M.). At least two sections were observed for determination of the existence of pale cells in tumor tissue according to the standard international protocol for pathological examination of RCCs [28,29]. ChRCCs were defined according to the 2016 WHO classification as tumors composed of large polygonal cells with reticular, clear or eosinophilic cytoplasm showing distinct cell borders, sometimes perinuclear halo and irregular (raisinoid) nuclei. 

Pale cells were described as being larger than eosinophilic cells, with voluminous pale, finely reticular, but not clear cytoplasm and with distinct cell borders. We used hematoxylin and eosin-stained sections and paid particular attention to the periphery of tumor cell sheet or nest, i.e., around the vascular septa and fibrous stroma in the tumor. 

### 4.2. The Cancer Genome Atlas (TCGA) Dataset

Clinical information of TCGA-KICH was obtained from the National Cancer Institute Genomic Data Commons Data Portal [30]. In TCGA-KICH dataset, there were 66 primary chRCCs with matched copy number variation data [13]. The demographic and clinical characteristics for the selected 66 patients are summarized in Table 1. Detailed clinical data can be found in Appendix A. For survival analysis, the patients with missing or with too short a follow-up (i.e., less than 30 days) were excluded from this study.

Digital whole slide images of TCGA cases were reviewed by using the Cancer Digital Slide Archive [21]. Publically available Level 3 TCGA data were downloaded from the FIREHOSE database [31], including GISTIC CN data. 

### 4.3. Japanese Patients

chRCCs with available histological material was retrieved from the archives of 13 of the authors’ institutions. The institutions are: Niigata University Medical & Dental Hospital (cases from 2002 to 2015), Kansai Medical University Hirakata Hospital (cases from 2007 to 2015), Seirei Hamamatsu General Hospital (cases from 2004 to 2015), Niigata Cancer Center Hospital (cases from 2002 to 2015), Tachikawa General Hospital (cases from 2007 to 2015), Gifu University Hospital (cases from 2005 to 2015), Niigata City General Hospital (cases from 2007 to 2015), Nagaoka Red Cross Hospital (cases from 2008 to 2015), Tohoku University Hospital (cases from 2008 to 2015), Nagasaki University Hospital (cases from 2014 to 2015), Iwate Medical University Hospital (cases from 2007 to 2015), Kochi Red Cross Hospital (cases from 2007 to 2015), and Aichi Medical University Hospital (cases from 2008 to 2015). The study did not include consultation cases. The study protocol was approved by the institutional review boards in all participating institutions. This study was a retrospective observational study, and an opt-out approach was used with the disclosure of this study on the website of each institution. The patients with missing or too short a follow-up (i.e., less than 30 days) were excluded from this study. All chRCCs were negative for vimentin except for focal sarcomatoid areas. The demographic and clinicopathological characteristics for the 119 chRCCs are summarized in Table 1.

### 4.4. OncoScan^®^ CNV Assay of chRCCs

Tumor areas displaying >80% cancer cells without hemorrhage or necrosis were marked on the hematoxylin and eosin slides. DNA from FFPE tumor tissue samples was obtained by punching 4 to 6 tissue cylinders (diameter 0.6 mm) from each sample. DNA extraction from FFPE tissue was done as described [32]. The double-strand DNA concentration (dsDNA) was determined using the fluorescence-based Qubit dsDNA HS Assay Kit (Thermo Fisher Scientific, Waltham, MA, USA). Tumors with poor DNA quality were excluded from the study. Genome-wide DNA copy-number alterations and allelic imbalances of 33 chRCC were determined using the Affymetrix OncoScan^®^ CNV Assay (Affymetrix/Thermo Fisher Scientific, Waltham, MA, USA) as previously described [33]. The demographic and clinicopathological characteristics for 33 Swiss chRCCs with clinical data are summarized in Table 1. Samples were further processed by IMGM Laboratories GmbH (Martinsried, Germany) for CNV (copy number variation) determination according to the Affymetrix OncoScan CNV Assay recommended protocol. The data were analyzed by the Nexus Copy Number 10.0 (Biodiscovery, Inc., El Segundo, CA, USA) software using Affymetrix TuScan algorithm. All array data were also manually reviewed for subtle alterations not automatically called by the software.

### 4.5. Statistical Analysis

All statistical analysis was done using R version 3.4.1 (R Foundation for Statistical Computing, Vienna, Austria) and EZR, Version 1.37 (Saitama Medical Center, Jichi Medical University, Saitama, Japan), which is a graphical user interface for R [34]. Fisher’s exact test was used to assess the association between two categorical variables. A Kaplan–Meier analysis and the log-rank test were used to derive and compare survival curves. Univariate and multivariate regression analyses with the Cox proportional hazards model were used to identify prognostic factors. The significance threshold was set at a *p*-value of 0.05.

## 5. Conclusions

In conclusion, the molecular difference between classic and eosinophilic chRCCs justifies the definition of 2 chRCC variants. Using the absence of pale cells as a diagnostic criterion for the eosinophilic variant may improve the reproducibility of histopathological subtyping.

## Figures and Tables

**Figure 1 cancers-11-01492-f001:**
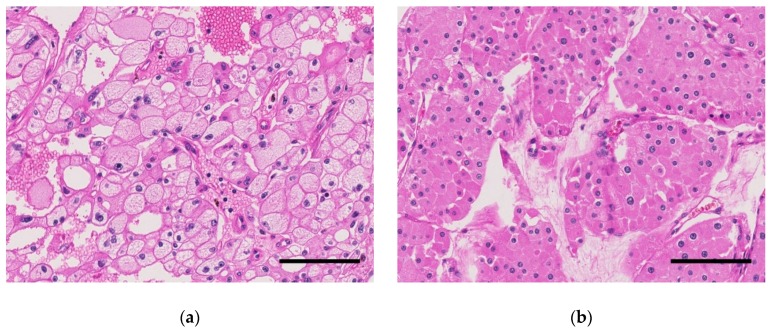
Representative microscopic images of chromophobe renal cell carcinomas (chRCCs) in the Swiss cohort (hematoxylin and eosin staining, scale bar, 100 µm). (**a**) classic chRCC with pale cells. (**b**) eosinophilic chRCC without pale cells.

**Figure 2 cancers-11-01492-f002:**
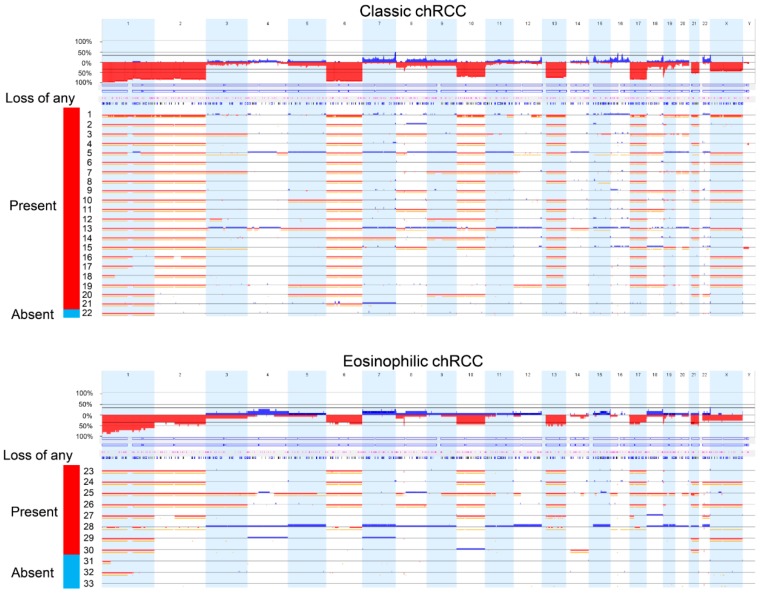
Copy number (CN) alterations and copy-neutral loss-of-heterozygosity detected by Affymetrix OncoScan^®^ CNV Assay of 22 classic chromophobe renal cell carcinomas (chRCCs) (upper panel) and 11 eosinophilic chRCCs (lower panel) in the Swiss cohort. Red signal, blue signal, and yellow signal show copy-number loss, copy-number gain, and copy-neutral loss-of-heterozygosity, respectively. Loss of any: Loss of any chromosome among chromosome 2, 6, 10, 13, 17, or 21. Red: Present, Blue: Absent.

**Figure 3 cancers-11-01492-f003:**
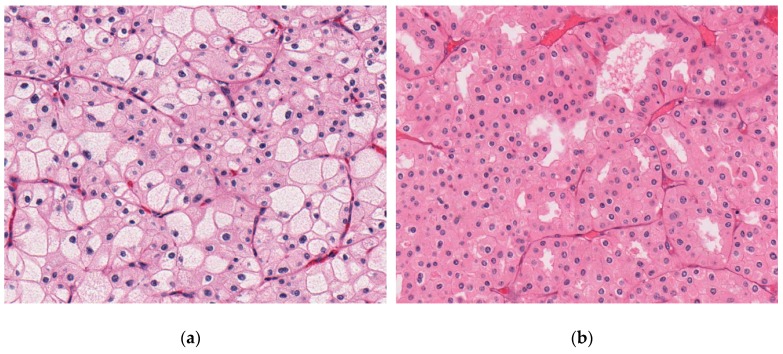
Representative microscopic images of chromophobe renal cell carcinomas (chRCCs) in the Cancer Genome Atlas Kidney Chromophobe (TCGA-KICH) cohort, whole-slide images from the Cancer Digital Slide Archive [21] (**a**) classic chRCC with pale cells (TCGA-KL-8345). (**b**) eosinophilic chRCC without pale cells (TCGA-KL-8326).

**Figure 4 cancers-11-01492-f004:**
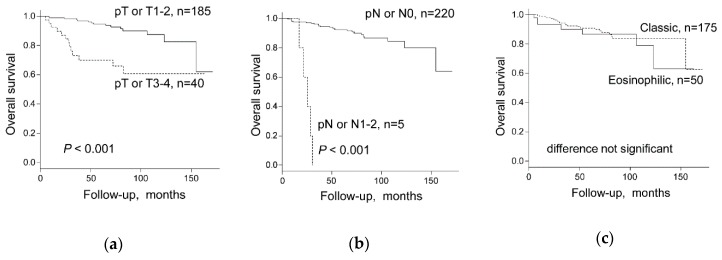
Overall survival stratified by (**a**) pT Stage or T stage (1–2 versus 3–4), (**b**) pN stage or N stage (pN1–2 versus pN0), and (**c**) chromophobe renal cell carcinoma (chRCC) subtype in all 225 chRCC patients combined from the Swiss, the Cancer Genome Atlas Kidney Chromophobe (TCGA-KICH), and Japanese cohorts. pT Stage or T stage: Swiss and Japanese cohorts, pT stage, and TCGA-KICH, T stage was used for the calculation. pN stage or N stage: Swiss and Japanese cohorts, pN stage, and TCGA-KICH, N stage was used for the calculation.

**Figure 5 cancers-11-01492-f005:**
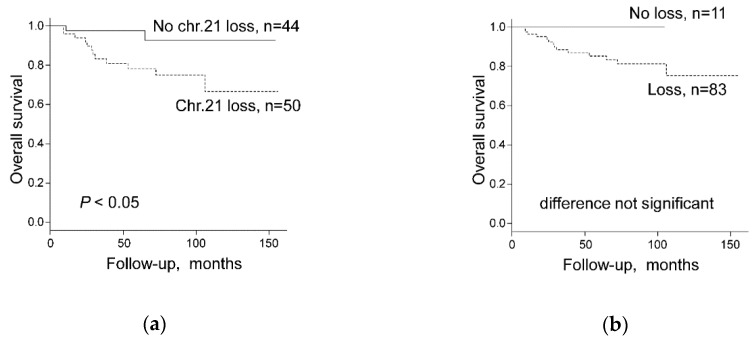
Overall survival stratified by (**a**) chromosome 21 status (Chr.21 loss versus No chr.21 loss) and (**b**) Loss of any chromosome among chromosome 1, 2, 6, 10, 13, 17, or 21 (Loss versus No loss) in 94 chRCC patients combined from Swiss and the Cancer Genome Atlas Kidney Chromophobe (TCGA-KICH) cohorts.

**Table 1 cancers-11-01492-t001:** Demographic and clinical characteristics.

Characteristics	Swiss Cohort (Total)	TCGA-KICH	Japanese Cohort
Patient number	42	66	119
Age (years)			
Range	18–87	17–86	26–88
Median	59	50	60
Gender			
Female	13 (31.0%)	27 (40.9%)	69 (58.0%)
Male	29 (69.0%)	39 (59.1%)	50 (42.0%)
pT Stage or T stage, *n* (%) *			
1	25 (59.5%)	21 (31.8%)	85 (71.4%)
2	11 (26.2%)	25 (37.9%)	19 (16.0%)
3	6 (14.3%)	18 (27.3%)	14 (11.8%)
4	0 (0%)	2 (3.0%)	1 (0.8%)
Subtype			
classic	32 (76.2%)	53 (80.3%)	90 (75.6%)
eosinophilic	10 (23.8%)	13 (19.7%)	29 (24.4%)

* Swiss and Japanese cohort: pT stage, TCGA-KICH: T stage.

**Table 2 cancers-11-01492-t002:** Copy number variation (chromosomal losses) in classic and eosinophilic chromophobe renal cell carcinomas (chRCCs) (Chr. = chromosome) from 33 Swiss chRCCs (Affymetrix OncoScan^®^ CNV Assay; Affymetrix/Thermo Fisher Scientific, Waltham, MA, USA) and combined Swiss/TCGA-KICH cohorts (The Cancer Genome Atlas copy number variation data).

Cohort	Swiss				Combined *			
Characteristics	n (%)	Classic chRCC ^a^ (%)	Eosinophilic chRCC ^b^ (%)	*p*-Value	n (%)	Classic chRCC ^a^ (%)	Eosinophilic chRCC ^b^ (%)	*p*-Value
Total	33	22 (66.7)	11 (33.3)		99	75 (75.8)	24 (24.2)	
Chr.1 status								
Loss	32 (97.0)	22 (100)	10 (90.9)	n.s.	87 (87.9)	70 (93.3)	17 (70.8)	<0.01
No loss	1 (3.0)	0 (0)	1 (9.1)		12 (12.1)	5 (6.7)	7 (29.2)	
Chr.2 status								
Loss	24 (72.7)	19 (86.4)	5 (45.5)	<0.05	73 (73.7)	63 (84.0)	10 (41.7)	<0.001
No loss	9 (27.3)	3 (13.6)	6 (54.5)		26 (26.3)	12 (16.0)	14 (58.3)	
Chr.6 status								
Loss	26 (78.8)	21 (95.5)	5 (45.5)	<0.01	78 (78.8)	68 (90.7)	10 (41.7)	<0.001
No loss	7 (21.2)	1 (4.5)	6 (54.5)		21 (21.2)	7 (9.3)	14 (58.3)	
Chr. 10 status								
Loss	21 (63.6)	16 (72.7)	5 (45.5)	n.s.	70 (70.7)	62 (82.7)	8 (33.3)	<0.001
No loss	12 (36.4)	6 (27.3)	6 (54.5)		29 (29.3)	13 (17.3)	16 (66.7)	
Chr.13 status								
Loss	23 (69.7)	17 (77.3)	6 (54.5)	n.s.	68 (68.7)	57 (76.0)	11 (45.8)	0.01
No loss	10 (30.3)	5 (22.7)	5 (45.5)		31 (31.3)	18 (24.0)	13 (54.2)	
Chr.17 status								
Loss	25 (75.8)	19 (86.4)	6 (54.5)	n.s.	75 (75.8)	64 (85.3)	11 (45.8)	<0.001
No loss	8 (24.2)	3 (13.6)	5 (45.5)		24 (24.2)	11 (14.7)	13 (54.2)	
Chr.21 status								
Loss	17 (51.5)	12 (54.5)	5 (45.5)	n.s.	52 (52.5)	42 (56.0)	10 (41.7)	n.s.
No loss	16 (48.5)	10 (45.5)	6 (54.5)		47 (47.5)	33 (44.0)	14 (58.3)	
Loss of any chromosome ^c^								
present	29 (87.9)	21 (95.5)	8 (72.7)	n.s.	83 (83.8)	69 (92.0)	14 (58.3)	<0.001
absent	4 (12.1)	1 (4.5)	3 (27.3)		16 (16.2)	6 (8.0)	10 (41.7)	

^a^ defined as the presence of pale cells, ^b^ defined as absence of pale cells, ^c^: Loss of any: Loss of any chromosome among chr. 2, 6, 10, 13, 17, or 21, n.s.: not significant, ***** combined Swiss/TCGA-KICH cohorts.

**Table 3 cancers-11-01492-t003:** Univariate and Multivariate Cox regression analysis on overall survival of 225 chRCC patients combined from Swiss, TCGA-KICH and Japanese cohorts.

Variables	Univariate		Multivariate	
	HR (95%CI)	*p*-Value	HR (95%CI)	*p*-Value
pT Stage or T stage * (3–4 vs. 1–2)	4.809 (2.275–10.16)	<0.001	3.177 (1.336–7.556)	<0.01
pN Stage or N stage * (1–2 vs. 0)	42.95 (13.16–140.1)	<0.001	21.140 (5.612–79.650)	<0.001
WHO/ISUP grade (Grade 3/4 vs. Grade 2) **	1.667 (0.502–5.537)	n.s.	2.010 (0.594–6.804)	n.s.
Subtype (classic vs. eosinophilic)	0.756 (0.321–1.782)	n.s.	0.520 (0.207–1.303)	n.s.

HR, hazard ratio, CI, confidence interval, n.s.: not significant. * Swiss and Japanese cohort: pT/pN stage, TCGA-KICH: T/N stage. ** WHO/ISUP grading system is not recommended for chRCC.

**Table 4 cancers-11-01492-t004:** Univariate survival analysis (Kaplan-Meier) with log-rank test (overall survival) of 94 chRCC patients combined from Swiss and TCGA-KICH cohorts (Chr. = chromosome).

Cohort	Swiss			Combined		
Characteristics	Cases, *n* (%)	Patient Death, *n* (%)	*p*-Value	Cases, *n* (%)	Patient Death, *n* (%)	*p*-Value
Total	30	5 (16.7)		94	14 (14.9)	
Chr.1 status						
Loss	30 (100)	5 (100)	n.s.	83 (88.3)	14 (100)	n.s.
No loss	0 (0)	0 (0)		11 (11.7)	0 (0)	
Chr.2 status						
Loss	22 (73.3)	4 (80.0)	n.s.	69 (73.4)	12 (85.7)	n.s.
No loss	8 (26.7)	1 (20.0)		25 (26.6)	2 (14.3)	
Chr.6 status						
Loss	25 (83.3)	4 (80.0)	n.s.	75 (79.8)	12 (85.7)	n.s.
No loss	5 (16.7)	1 (20.0)		19 (20.2)	2 (14.3)	
Chr. 10 status						
Loss	19 (63.3)	3 (60.0)	n.s.	67 (71.3)	12 (85.7)	n.s.
No loss	11 (36.7)	2 (40.0)		27 (28.7)	2 (14.3)	
Chr.13 status						
Loss	21 (70.0)	4 (80.0)	n.s.	64 (68.1)	11 (78.6)	n.s.
No loss	9 (30.0)	1 (20.0)		30 (31.9)	3 (21.4)	
Chr.17 status						
Loss	23 (76.7)	4 (80.0)	n.s.	71 (75.5)	11 (78.6)	n.s.
No loss	7 (23.3)	1 (20.0)		23 (24.5)	3 (21.4)	
Chr.21 status						
Loss	17 (56.7)	4 (80.0)	n.s.	50 (53.2)	12 (85.7)	<0.05
No loss	13 (43.3)	1 (20.0)		44 (46.8)	2 (14.3)	
Loss of any chromosome *						
present	30 (100)	5 (100)	n.s.	83 (88.3)	14 (100)	n.s.
absent	0 (0)	0 (0)		11 (11.7)	0 (0)	

* Loss of any chromosome: Loss of any chromosome among chr. 1, 2, 6, 10, 13, 17 or 21, n.s.: not significant.

**Table 5 cancers-11-01492-t005:** Univariate and Multivariate Cox regression analysis (overall survival) of 94 chRCC patients combined from Swiss and TCGA-KICH cohorts.

Variables	Univariate		Multivariate		Multivariate	
	HR (95%CI)	*p*-Value	HR (95%CI)	*p*-Value	HR (95%CI)	*p*-Value
pT Stage or T stage ^a^ (3–4 vs. 1–2)	6.323 (2.078–19.25)	0.001	5.505 (1.935–17.450)	<0.01	6.344 (2.010–20.03)	<0.01
Loss of any chromosome ^b,c^ (present vs. absent)	2.503 (0.331–320.637)	n.s.	1.319 (0.154–172.711)	n.s.		
Loss of chromosome 21 (present vs. absent)	4.684 (1.047–20.95)	<0.05			4.480 (1.000–20.08)	n.s.

HR, hazard ratio, CI, confidence interval, n.s.: not significant. a: Swiss and Japanese cohort: pT stage, ^a^ Swiss and Japanese cohort: pT stage, TCGA-KICH: T stage, ^b^ Loss of any: Loss of any chromosome of chromosome 1, 2, 6, 10, 13, 17, or 21, ^c^ Firth correction was used because of quasi-complete separation, there was no event in one of the subgroups.

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
