# Peer review of "Classic Chromophobe Renal Cell Carcinoma Incur a Larger Number of Chromosomal Losses Than Seen in the Eosinophilic Subtype"

_cancers, 2019, doi:10.3390/cancers11101492_

Round 1

Reviewer 1 Report

This is a retrospective study of comparing the chromosomal copy number variation in total 99 patients (75 classic chRCC and 24 eosinophilic chRCC). Authors found that losses of chromosome 1, 2, 6, 10, 13 and 17 were significantly more frequent in classic chRC. However, there was no survival difference between two groups in total 225 patients. This study may have limited interest to the clinician, because it is unclear what meaning does this chromosomal copy number variation difference have. Author must address the following criticisms listed as below.

Overall numbers are small, and in particular, the number of eosinophilic chRCC is only 24. On this basis, it is not clear what the clinical implications are of the study findings. To clarify the clinical meaning of losses of chromosome, the authors should evaluate the survival with each chromosomal copy number variation status. pT stage include the tumor size. So, must not analyze both factors at the same time. The authors should discuss a reasonable theory as to why / how chromosomal copy number variation status result in the different phenotype in chRCC. The discussion is very brief, and would benefit from a deeper discussion on the literature on the topic.

Reviewer 2 Report

Paper written by Brunelli et al.(Mod Pathol. 2005Feb;18(2):161-9) showed losses of chromosomes 1, 2, 6, 10, and 17 are frequent in both eosinophilic and classic chromophobe renal cell carcinomas.In the present study combined Swiss and TCGA-KICH cohorts, losses of chromosome 1, 2, 6, 10, 13 and
17 were significantly more frequent in classic chRCC ,suggesting that classic chRCC are characterized by higher chromosomal instability.

Its just and incremental work reaffirming previous observation. It will be more useful if authors can discuss the relation between this sort of genetic instability and therapeutic outcome.

Round 2

Reviewer 1 Report

Nice revision for the revised comments. These data seem acceptable.